# Call for Caution to Consider *Babesia bovis* and *Babesia bigemina* as Anthropozoonotic Agents in Colombia. Comment on Kumar et al. The Global Emergence of Human Babesiosis. *Pathogens* 2021, *10*, 1447

**DOI:** 10.3390/pathogens11020263

**Published:** 2022-02-18

**Authors:** Carlos Ramiro Silva-Ramos, Álvaro A. Faccini-Martínez

**Affiliations:** 1Grupo de Enfermedades Infecciosas, Departamento de Microbiología, Facultad de Ciencias, Pontificia Universidad Javeriana, Bogotá 110221, Colombia; ramcar007@gmail.com; 2Department of Pathology, University of Texas Medical Branch, Galveston, TX 77550, USA; 3Committee of Tropical Medicine, Zoonoses and Travel Medicine, Asociación Colombiana de Infectología, Bogotá 110221, Colombia

**Keywords:** human babesiosis, *Babesia bovis*, *Babesia bigemina*, Colombia

## Abstract

Currently, six species and two genetic variants within *Babesia* genus have been confirmed as human pathogens. *Babesia bovis* and *Babesia bigemina* are causative agents of bovine babesiosis, and, in spite of the worldwide distribution of those species and their vectors, no description of related human cases has been reported. As a contribution, we would like to address the articles which claim the alleged role of *B. bovis* and *B. bigemina* as anthropozoonotic pathogens in Colombia.

We have read with great interest the review by Kumar et al., “The Global Emergence of Human Babesiosis”, published in *Pathogens*, in which several aspects on the epidemiology of this tick-borne disease are addressed, including the geographic distribution of reported cases and related *Babesia* species [1].

In Table 2 of their review, Kumar et al. mention the causative agents of human babesiosis throughout the world, designating *Babesia bovis* and *Babesia bigemina* as associated with human cases in Colombia, South America [1]. Nevertheless, in the Table’s footnote and “Human babesiosis in the Americas” subheading, they highlight that “some causative agents have not been confirmed in larger case series so are not yet accepted as established causes of human babesiosis” and “human babesiosis due to *B. bovis* and *B. bigemina* had not previously been described”, respectively. In line with the above statements, we would like to address publications which claim the alleged role of these *Babesia* species as anthropozoonotic pathogens in Colombia [2,3].

Currently, among the more than the 100 *Babesia* spp. identified in wild and domestic animals worldwide, six species (*Babesia microti*, *Babesia divergens*, *Babesia duncani*, *Babesia venatorum*, *Babesia motasi*, and *Babesia crassa*-like pathogen) and two genetic variants (*Babesia divergens*-like and *Babesia microti*-like) have been confirmed as human pathogens [1,4]. Related vectors are anthropophilic ixodid ticks from the temperate regions of the northern hemisphere, in the *Ixodes ricinus* complex (*Ixodes persulcatus*, *Ixodes ricinus*, *Ixodes scapularis*), as well as *Dermacentor albipictus* and *Ixodes ovatus* [1,4]. In contrast, *B. bovis* and *B. bigemina* are causative agents of bovine babesiosis, which affect cattle and buffaloes, causing severe disease of considerable economic impact due to loss of meat production and death of infected animals [5]. These bovine babesiae are transmitted by *Rhipicephalus* (*Boophilus*) tick species, which are widespread in tropical and subtropical regions [6].

Curiously, in spite of the worldwide distribution of *B. bovis*, *B. bigemina*, and their vectors [5,6], no description of related human cases have been reported, beyond the Colombian studies by Ríos et al. [2] and González et al. [3]. The former described seven farm workers, with or without malaria-like symptoms, from Puerto Berrío municipality (Antioquia department) with IgM or IgG antibodies to *Babesia* sp. by IFA test, using *B. bovis* and *B. bigemina* antigens [2]. All but one had negative IgM/IgG antibodies to *Plasmodium falciparum* by IFA and ELISA tests [2]. One of the symptomatic individuals, in addition to an IgM titer of 64 to *B. bovis*, presented pyriform parasites in pairs and tetrads without pigment in a thin blood smear, suggesting *Babesia* infection [2].

While this patient could be a probable babesiosis case considering the blood smear finding, a positive serological result to *B. bovis* does not necessarily indicate a specific infection with this *Babesia*, but rather, an exposure to an unknown *Babesia* sp. or even a cross-reaction [7,8]. Indeed, despite the fact that Puerto Berrío municipality is an endemic area for *P. falciparum* and *Plasmodium vivax* malaria [9], Ríos et al. did not include this latter *Plasmodium* species in order to rule out false positive results by *B. bovis* and *B. bigemina* IFA testing [2]. In this sense, interestingly, in 1972, Ludford et al. demonstrated that 3/20 individuals throughout the course of an induced *P. vivax* infection developed antibodies to *B. bovis* of equal or higher titer than those to *P. vivax*, and lower or negative titers to *P. falciparum* [10].

Regarding the other Colombian study, González et al., using molecular, microscopic and serological methods, investigated *B. bovis* and *B. bigemina* infection in 300 humans involved in cattle raising from Turbo and Necoclí municipalities, Antioquia department [3]. Overall, four (1.3%) and two (0.6%) individuals studied for possible infection with *B. bovis* and *B. bigemina*, respectively, were detected by PCR of venous blood samples [3]. In peripheral blood smears, parasitic forms suggesting *Babesia* spp. were observed in two individuals from the *B. bovis* positive-PCR group and one individual from the *B. bigemina* positive-PCR group [3]. In addition, the authors reported detection of antibodies for both bovine babesiae in one subject (0.3%), by ELISA and IFA tests, using *B. bovis* and *B. bigemina* antigens [3].

Similarly, as in the Rios et al. study, González et al. described Turbo and Necoclí municipalities as malaria endemic areas [3]; nevertheless, they did not rule out potential serological cross-reaction between *B. bovis*/*B. bigemina* and *Plasmodium* spp. [10]. Moreover, regarding molecular detection, González et al. used a species-specific nested-PCR, with expected products of 291 bp for *B. bovis* and 178 bp for *B. bigemina* of the 18S gene, and subsequent sequencing [3]. The authors did not mention the positive controls that we assume were *B. bovis*/*B. bigemina* DNA. Thus, it is well known that nested PCR testing is generally more prone to false positive results [11], and unfortunately, the manuscript does not provide information on generated sequences of the six positive human samples, which would have been valuable in order to confirm that it was not the result of contamination. Regarding this, even excluding contamination, humans could be participating as merely accidental hosts rather than playing a role in the parasite life cycle itself, as *B. bovis* and *B. bigemina* might be able to undergo a few replication cycles in the human blood, but then die [12].

Due to the worldwide emergence of some infectious diseases, the presence of human pathogenic *Babesia* spp. in Colombia must not be totally discarded. However, the epidemiology of human babesiosis is not different from other vector-borne diseases, which are highly related to the geographic distribution of their arthropod vectors [1]. Thus, as mentioned, for human pathogenic babesiae, the vectors are anthropophilic ixodid ticks from the temperate regions of the northern hemisphere, and none of them have been found in tropical regions, including Colombia [13,14]. In addition, it is noteworthy that *R. microplus*, the main vector of *B. bovis* and *B. bigemina* in the Neotropical Region [6], is a sporadic anthropophilic tick [15].

Finally, and interestingly, confirmed cases of human babesiosis due to *B. microti* or *B. microti*-like protozoa have been reported in South America (Bolivia and Ecuador) [16,17] and in Mexico [18]. Thus, in addition to the well-known cross-reactivity which occurs between *B. microti* and *B. bovis*/*B. bigemina* [7,8], the fact that *B. microti* or a phylogenetically related species has been detected in human cases in these geographic regions implies that at least one zoonotic *Babesia* sp. is circulating in tropical countries, generating the question of what the tick vector is; an issue deserving further studies. However, according to the currently available worldwide data, the claim of *B. bovis* and *B. bigemina* as etiological agents of human babesiosis should not yet be considered until further scientific evidence demonstrates their potential as anthropozoonotic pathogens.

## Data Availability

Not applicable.

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
