# Peer review of "Call for Caution to Consider Babesia bovis and Babesia bigemina as Anthropozoonotic Agents in Colombia. Comment on Kumar et al. The Global Emergence of Human Babesiosis. Pathogens 2021, 10, 1447"

_pathogens, 2022, doi:10.3390/pathogens11020263_

Round 1

Reviewer 1 Report

The ms is of interest to readers and serves to analyze previous publications on the subject. 

Reviewer 2 Report

Authors properly addressed the concerns of the reviewers.

This manuscript is a resubmission of an earlier submission. The following is a list of the peer review reports and author responses from that submission.

Round 1

Reviewer 1 Report

This is a "comment" article discussing the likelihood of Babesia bovis and B. bigemina as potential agents of human babesiosis. The article attempts clarifying some points in a previous article by Kumar et al. “The global emergence 19 of human babesiosis”, Pathogens 202110(11), 1447) that includes these two agents as possible agents of human babesiosis in Colombia, based on two published reports.  However, the article by Kumar et al., also highlight the fact that these cases require further confirmation and that B. bovis and B. bigemina have not been described previously as agents of human babesiosis. The impression of this reviewer is that the article by Kumar et al., correctly describes that situation and reference the studies for further evaluation by the readers, and feel that no further clarification is needed on this issue. However, the authors in the comment article pinpoint deficiencies in the two references (references  2 and 3 in the article) describing human babesiosis in Colombia and the fact that the common vectors of bovine babesiosis (Rhipicephalus spp.) are not likely to feed in humans, and correctly suggest that the identification of  possible tick(s) vector(s) involved in these cases is also required. However, it appears from the last paragraph of the article that the authors speculate or  imply that a B. microti, or a B. microti-like parasite might be ultimately responsible for the cases of human babesiosis in Colombia. If that is the case, authors should also comment on whether there is any described evidence of serological and PCR cross-reactivity between B. microti and the bovine babesiosis agents. It appears unlikely, since B. microti is not considered as a sensu stricto Babesia and antigenically distinct from the bovine babesiosis agents.  An important mistake in line 99 in this article is that the authors consider B. microti or B. microti -like as “bacteria”, and that needs to be corrected.   

In summary, the impression of this reviewer is that this article does not add much to what  is already described the article by Kumar et al.

A few mistakes needs to be addressed:

Line 40: add “In”. so it reads “In contrast….”

Line 73; please re-write this sentence it is not clear as written.

Line 86 : replace “certain” by “confirm”.

 Line 99: Please remove “bacteria”, since Babesia are considered protozoan apicomplexan parasites.

Reviewer 2 Report

The authors present very clearly a revision of a recent article published in Pathogens, related to consider Babesia bovis and B. bigemina as zoonotic. I agree with the authors that there is not enough scientific evidence to accept this hypothesis.  In addition to the presented arguments, the authors should also include the possibility of humans as accidental hosts as has been observed for several Babesia spp. In these accidental infections, the parasite might be able to undergo a few replication cycles in the blood but then dies, and thus these accidental hosts do not play a role in the parasite epidemiology. Please check and cite the following publication where the subject of accidental hosts is dealt with:

Schnittger L, Rodriguez AE, Florin-Christensen M, Morrison DA. Babesia: a world emerging. Infect Genet Evol. 2012 Dec;12(8):1788-809. doi: 10.1016/j.meegid.2012.07.004.

The article is  well written. 

Line 94: "it" is noteworthy...

Line 99: change "Babesia-like bacteria" for "Babesia-like protozoa" or Babesia-like piroplasmdis". Babesia sp. are eukaryotic organisms, not bacteria.